# Effects of Medication Period and Gastrin Levels on Endoscopic Gastric Mucosal Changes in Long-Term Proton Pump Inhibitor Users

**DOI:** 10.3390/diagnostics14222540

**Published:** 2024-11-13

**Authors:** Hiroko Suda, Koichi Sakurai, Sachi Eto, Satomi Fujie, Ayako Okuda, Takayuki Takeichi, Masayuki Urata, Tetsuya Murao, Kiwamu Hasuda, Masahiro Hirano, Yo Kato, Ken Haruma

**Affiliations:** 1Hattori Clinic, 2-12-35 Shin-Machi, Chuo-ku, Kumamoto City 860-0004, Kumamoto, Japan; sachi.666666@gmail.com (S.E.); satomi-fujie@hotmail.co.jp (S.F.); ayako19770422@gmail.com (A.O.); ttake@wg7.so-net.ne.jp (T.T.); urajirou@ina.bbiq.jp (M.U.); te-mura@hattori-clinic.com (T.M.); ki-hasu@hattori-clinic.com (K.H.); 2Hirano Gastroenterology Clinic, 2-3029-2 Onuki-cho, Nobeoka 882-0803, Miyazaki, Japan; gihirano@gaea.ocn.ne.jp; 3Hibiya Digital Diagnostic Pathology Clinic, 2-2-3 Uchisaiwai-cho, Chiyoda-ku, Tokyo 100-0011, Japan; dr.yokato@gmail.com; 4Department of Internal Medicine 2, Kawasaki Medical School General Medical Center, 577 Matsushima, Kurashiki-City 701-0192, Okayama, Japan

**Keywords:** proton pump inhibitors, long-term medication period, gastrin, PPI-related endoscopic gastric mucosal changes, cracked/cobblestone-like mucosa

## Abstract

**Background/Objectives**: Proton pump inhibitor (PPI) use has increased worldwide, including in continuous and longer-term users. Recent reports highlight PPI-related endoscopic gastric mucosal changes, including fundic gland polyps, hyperplastic polyps, multiple white and flat elevated lesions, cracked and cobblestone-like mucosa (CCLM), and black spots. PPI use elevates gastrin levels because of acid inhibition, and hypergastrinemia might be relevant to these findings. In this cross-sectional study, we retrospectively examined gastric mucosal changes in long-term PPI users, focusing on medication period and gastrin levels. **Methods**: We enrolled 57 patients who received a PPI (>1 year) at two clinics between January 2021 and March 2022. Participants were classified according to medication period: 1 < 5, 5–10, and ≥10 years. Gastrin levels were categorized as low, middle, and high (<250, 250–500, and ≥500 pg/mL, respectively). Odds ratios (OR) were estimated to assess the risk of endoscopic findings. **Results**: Of the 57 patients, 6 (10.5%), 25 (43.9%), and 26 (45.6%) were PPI users of 1 < 5, 5–10, and ≥10 years, respectively. There were no significant differences in the incidence of endoscopic findings among the medication periods. Low, middle, and high gastrin groups included 21 (36.8%), 21 (36.8%), and 15 (26.3%) patients, respectively. CCLM incidence was significantly elevated in higher gastrin level groups: middle (OR, 6.60; 95% confidence interval [CI], 1.46–29.75; *p* = 0.014) and high (OR, 9.00; 95% CI, 1.79–45.23; *p* = 0.0008) (p-trend = 0.0171). No significant differences were observed for other findings. **Conclusions**: No elevated risk of PPI-related gastric epithelial changes in long-term PPI users was observed time-dependently. Notably, higher gastrin levels were positively associated with CCLM development, irrespective of the medication period.

## 1. Introduction

Since the introduction of omeprazole in 1989, proton pump inhibitors (PPIs) have become clinically essential. Instead of histamine type 2 receptor antagonists, PPIs are used to treat acid-related diseases, such as gastroesophageal reflux and peptic ulcer disease, and are also prescribed as a prophylactic strategy for aspirin-induced ulcers [1,2]. PPIs have been widely prescribed for >30 years.

Distinctive endoscopic gastric mucosal changes have been reported in patients using PPIs. Since Graham reported three cases of fundic gland polyps (FGPs) that developed after 1 year of omeprazole therapy in 1992 [3], hyperplastic polyps, multiple white and flat elevated lesions (MWFL), cracked and cobblestone-like mucosa (CCLM), and black spots have been reported as PPI-associated endoscopic findings [4,5,6]. Histologically, these endoscopic findings are characterized by gastric epithelial changes such as parietal cell enlargement, cystic dilation of the fundic glands, and foveolar epithelial hyperplasia [4,7,8].

In patients using PPIs, hypergastrinemia is a well-known condition hormonally induced by acid inhibition. Gastrin plays an important role in stimulating acid secretion and regulating the overall balanced growth of the oxyntic mucosa. The gastrin receptor (CCK2) is predominantly found on parietal cells and enterochromaffin-like (ECL) cells in the human stomach [9,10,11]. On the other hand, the presence of gastric receptors in ECL cells raises concerns that conditions such as autoimmune gastritis (AIG) [12,13], which presents with chronic hypergastrinemia, may stimulate ECL cells to hyperplasia and further endocrine cell micronest (ECM) formation, resulting in carcinoid development [14]. Recent evidence also suggested that CCK2 receptors are also expressed on progenitor cells located in the proliferative zone [15]. Gastrin stimulates the proliferation of these cells and exerts a growth factor effect through the paracrine action of other growth factors, such as the epidermal growth factor family members. Although our understanding of the physiological mechanisms remains limited, gastrin is probably involved in regulating the development of the gastric epithelium [9,11,16]. Therefore, PPI-related endoscopic gastric mucosal changes are likely affected by hypergastrinemia.

PPI use has increased worldwide. Although the negative effects of long-term PPI use due to hypochlorhydria/achlorhydria or dysbiosis have been a concern, the therapeutic benefits of PPIs outweigh the risks of such side effects [17,18]. In clinical practice, PPI use has increased over time, including in continuous and longer-term users [19,20,21,22], which raises concerns about the risks of long-term PPI use. Recently, some studies have reported PPI-related gastric mucosal changes in patients using PPIs; however, only a very few studies have included patients receiving PPIs for the long term (>10 years), and the involvement of hypergastrinemia is unclear [5,23]

Therefore, this cross-sectional study of endoscopic gastric mucosal changes in patients taking PPIs for the long term, including those with a medication history exceeding 10 years, aimed to assess the effects of the medication period and gastrin levels on gastric epithelial changes.

## 2. Materials and Methods

### 2.1. Study Design and Data Collection

We recruited patients who received PPI maintenance treatment for gastroesophageal reflux symptoms for >1 year between January 2021 and March 2022 at the Hattori and Hirano Clinics. The PPI agents used were omeprazole, rabeprazole, lansoprazole, and esomeprazole. Data on patients’ history of PPI use were collected from patients’ medical records. Patients with insufficient data regarding the history of PPI use in the prescription record, poor drug compliance <90%, gastric surgery, any comorbidities that precluded safe participation in this study, or pregnancy were excluded.

### 2.2. Endoscopic Evaluation

Participants underwent routine esophagogastroduodenoscopy (EGD) for abdominal symptoms, particularly heartburn and regurgitation. All examinations were performed using GIF-H290Z with an EVIS LUCERA ELITE system (Olympus Medical Systems, Tokyo, Japan). EGD was performed by nine endoscopists with >7 years of experience. Atrophic gastritis was defined according to the Kimura–Takemoto classification [24]. Endoscopic intestinal metaplasia was evaluated using the Kyoto classification. The absence of intestinal metaplasia was scored as 0, the presence of intestinal metaplasia within the antrum was scored as 1, and intestinal metaplasia extending into the corps was scored as 2 [25]. Three endoscopists retrospectively evaluated the presence of gastric mucosal changes associated with PPIs, including CCLM, MWFL, black spots, FGPs, and hyperplastic polyps (Figure 1). CCLM included cracked and/or cobblestone-like mucosa. The MWFL was defined as a sharply demarcated white and slightly elevated mucosa. Black spots were indicated as punctate black pigments. FGPs and hyperplastic polyps were evaluated endoscopically without histological evaluation. Other endoscopic findings, including tumor lesions, were also reviewed.

### 2.3. Blood Samples

Fasting blood samples, collected >12 h after the last meal, were collected from participants shortly before EGD. Biochemical tests were performed, and serum gastrin levels were measured using radioimmunoassay (normal range, <200 pg/mL) (SRL., Tokyo, Japan).

### 2.4. H. pylori Infection Status Evaluation

*Helicobacter. pylori* (*H. pylori*) infection was diagnosed based on the presence of mucosal atrophy on endoscopic images and positive results for serum anti-*H. pylori* immunoglobulin G antibody test (>3 U/mL) and urea breath test (>2.5‰). A history of *H. pylori* eradication in the medical records was also assessed.

### 2.5. Histopathological Evaluation

The gastric biopsy for histopathological evaluation of the gastric mucosa was performed at the time of index EGD. The sample was collected from the fundic gland region of the upper-greater curvature of the stomach. A pathologist specialized in gastrointestinal histopathology evaluated the presence of endocrine cell micronest (ECM) and parietal cell hyperplasia.

### 2.6. Definitions for Chronological Assessment and Gastrin Levels

For chronological assessment, the participants were classified according to medication period at the time of index EGD: ≥1, ≥5, and ≥ 10 years. A previous Japanese study reported a mean gastrin level of 95.9 ± 74.6 pg/mL (mean ± SD), with 95.9 + 2SD calculated as 245.1 pg/mL, approximately 250 pg/mL. Consequently, a threshold of 250 pg/mL was established to define hypergastrinemia [26,27]. Based on this definition, the threshold for the middle gastrin group was set at 250 pg/mL, while the threshold for the high gastrin group was defined as 500 pg/mL, which is double the threshold for hypergastrinemia. Following this definition, participants were categorized based on gastrin levels, <250 pg/mL (low), ≥250 pg/mL, <500 pg/mL (middle), and ≥500 pg/mL (high), to clearly understand the distribution of gastrin levels within participants. Baseline characteristics, blood samples, and endoscopic findings were compared among the groups.

### 2.7. Statistical Analysis

Statistical analysis was performed with EZR (Saitama Medical Center, Jichi Medical University, Saitama, Japan) and STATA 18MP (StataCorp LLC, College Station, TX, USA). Continuous variables were expressed as medians and quartiles. The Kruskal–Wallis test for medians was used to compare the groups. Categorical variables were expressed as numbers (percentages) and compared using Fisher’s exact test. Odds ratios (OR) were estimated using a logistic regression model with robust variance. Statistical significance was set at *p* < 0.05.

## 3. Results

### 3.1. Demographic and Clinical Characteristics

A total of 57 Japanese patients were enrolled in this study. The PPI agents used were esomeprazole (*n* = 40), omeprazole (*n* = 10), lansoprazole (*n* = 6), and rabeprazole (*n* = 1). The median (interquartile range [IQR]) age was 73 (68–77) years. Among the participants, 22 (38.6%) were male, 6 (10.5%) were current smokers, and 18 (31.6%) were alcohol drinkers. Of the 57 patients, 26 (45.6%) had been taking a PPI for >10 years. Regarding H. pylori infection status, 29 (50.8%) patients were not infected, 19 (33.3%) received post-eradication therapy, and 6 (10.5%) had current infections. The infection status of three patients was undetermined because of insufficient data in the clinical records on the history of H. pylori eradication at previous hospitals. A total of 33 (57.9%) patients showed no atrophic gastritis, whereas 24 (42.1%) patients showed atrophic gastritis. A total of 47 (82.4%) patients showed no intestinal metaplasia. The median (IQR) gastrin level was 310 (190–510) pg/mL. The overall incidences of CCLM, MWFL, and black spots were 23 (40.4%), 10 (17.5%), and 4 (7.0%), respectively. The overall incidence of FGPs and hyperplastic polyps was 34 (59.6%) and 13 (17.5%), respectively (Table 1).

### 3.2. Relationship Between Endoscopic Findings and the PPI Medication Period

The baseline demographic and clinical characteristics categorized by the PPI medication period were not statistically significantly different (Table 2). There was no significant difference in the incidence of all endoscopic findings among the treatment periods. (Figure 2, Table 3). The OR could not be calculated due to the small number of patients with MWFL.

### 3.3. Relationship Between Endoscopic Findings and Gastrin Level

There were no significant differences in the baseline characteristics except alcohol intake among the three groups stratified by gastrin levels (Table 4). The incidence of CCLM was significantly higher in the middle- and high-gastrin level groups (Figure 2). As shown in Table 5, according to the OR calculation, the patients in higher gastrin level groups had a higher incidence (middle-gastrin level group [OR, 6.60; 95% confidence interval (CI), 1.46–29.75; *p* = 0.014] and high-gastrin group [OR, 9.00; 95% CI, 1.79–45.23; *p* = 0.008]), indicating significantly increased odds in these groups compared with the low-gastrin level group (p-trend = 0.0171). The relationship remained significant in the bivariate analysis adjusted for age, sex, medication period, *H. pylori* infection status, atrophic pattern, and intestinal metaplasia (Table 6).

Notably, histopathological analysis revealed that parietal cell hyperplasia was more frequently observed in the high-gastrin group, while ECM distribution showed no significant differences (Table 4). Additionally, no patients exhibited findings indicative of AIG.

## 4. Discussion

This observational study investigated the effect of long-term PPI use on gastric epithelial mucosa from two perspectives, the medication period and gastrin level. The treatment period did not affect the PPI-related endoscopic features. Higher gastrin levels significantly increased CCLM risk, irrespective of the medication period.

The definitions of long-term PPI use vary among studies due to the lack of consensus [28]. Some studies have reported PPI-related gastric mucosal changes in patients receiving PPIs in the long term; however, no studies have investigated the risk of gastric mucosal changes associated with medication periods encompassing PPI use for over a decade. In the present study, we compared endoscopic findings related to PPI use in 1–5, 5–10, and >10 years. Our results showed no significant differences in any of the endoscopic findings among the medication periods. Patients who received a PPI for a longer period showed an increased risk of CCLM; however, no significant difference was observed. Jalving et al. conducted a study assessing the risk of FGPs in patients using a PPI for <1, 1–5, and >5 years. Their findings suggested a continuous increase in the risk of FGPs over time (<1 year: OR 1.0, 95% CI: 0.5–1.8; 1–4.9 years: OR 2.2, 95% CI: 1.3–3.8; >5 years: OR 3.8, 95% CI: 2.2–6.7) [7]. In the present study, no significant differences in the incidence of FGPs were observed between patients who received a PPI for 1–5 and >5 years. This discrepancy could potentially be attributed to advancements in endoscopic technology, which provides a higher resolution that facilitates the detection of very small FGPs. Another factor to consider is the variation in the PPI classes. Jalving et al. did not specify the PPI class, whereas approximately 70% of the participants in the present study used esomeprazole. The results of the present study suggest that the risk of PPI-related endoscopic features in patients who have received a PPI for >10 years did not increase, implying that these endoscopic features were established within the first 10 years of PPI use and did not progress thereafter.

We observed an association between CCLM and gastrin levels in patients using PPIs. Elevated gastrin levels in patients receiving PPIs in the long term were positively associated with CCLM development. In contrast, Miyamoto et al. reported no association between gastrin levels and CCLM [5]. This discrepancy may be because their analysis considered cracked mucosa and cobblestone-like mucosa as different features. In the present study, we combined both features into one endoscopic category, considering that they arise from a common mechanism with differences in the extent of oxyntic gland dilatations. Hongo et al. reported that patients using a PPI presenting with FGP did not have high gastrin levels (approximately 200 pg/mL); however, high gastrin levels (>400 pg/mL) significantly increased the risk of hyperplastic polyps (Hazard ratio = 4.923; 95% CI, 1.486–16.31) [29]. In the present study, although the difference was not statistically significant, there was a trend suggesting that FGPs were less prevalent in the high gastrin level group, whereas hyperplastic polyps were less prevalent in the low gastrin level group. Gastrin levels may also influence the development of stomach polypoid lesions. From a histological perspective, CCLM and FGPs are characterized by parietal cell protrusion, parietal cell hyperplasia, and cystic dilatation of the fundic glands [5,7,23]. Parietal cell changes are a characteristic feature in patients using a PPI [30]. Cystic dilatation of the oxyntic glands is believed to be caused by fluid collection due to outflow obstruction of the gland from parietal cell proliferation and hyperplasia [30,31]. A study using gastrin-deficient mice reported that gastrin stimulates the movement of parietal cells along the gland axis [32]. In addition, other studies have reported that gastrin regulates the expression of KCNQ1 and AQP4 on parietal cells [16,33], which play a role in acid secretion and water transport. These studies suggest that PPI-related endoscopic features may develop intricately, intertwined with the proliferation rate of parietal cells and the extent of fluid collection affected by gastrin levels. Considering our histological findings, which indicated that ECMs were not affected by gastrin levels, while parietal cell hyperplasia was more pronounced in the high gastrin group, we speculated that the primary mechanism of CCLM formation is the proliferation of parietal cells accelerated by hypergastrinemia, rather than the proliferation of ECL cells. This is the first study to suggest an association between PPI-related endoscopic findings and hypergastrinemia. Further research is required to understand how gastrin affects gastric epithelial cells in a dose-dependent manner.

Tumor development caused by hypergastrinemia due to chronic PPI use has been a concern [34]. A family with an inherited defect in the H+K+ ATPase gastric proton pump gene (mutation in *ATP4A* gene), who could not secrete gastric acid and consequently had hypergastrinemia from birth, developed gastric carcinoids in their third or fourth decade [35]. This indicates the possibility of carcinoid tumor development in patients using PPIs for a much longer term. In the present study, no patient developed gastric carcinoids or any other neoplastic lesions, and no correlation was found between the endoscopic findings and the PPI medication period. These findings indicated that long-term PPI use seems to be safe with regard to tumor development. However, given that the proportion of patients using PPIs for the long term, including younger generations, is increasing, and a more intensive anti-acid drug, P-CAB, has been developed [36], more attention should be paid to the risk of tumor development due to hypergastrinemia. The endoscopic features observed in this study could potentially serve as indicators of gastrin levels in patients using PPI therapy. Specifically, the presence of CCLM might be a useful marker for identifying patients with severe hypergastrinemia. Although our findings did not show an association between ECM distribution and either the duration of medication use or gastrin levels, it is worth noting that severe hypergastrinemia, such as that seen in AIG, has been linked to an increased risk of gastric tumorigenesis [13]. The behavior of the parietal proliferation after long-time PPI use, which is not observed in AIG patients, remains unknown. In patients with CCLM, minimizing anti-acid use may be advisable. However, further research is necessary to elucidate the long-term consequences of hypergastrinemia in patients using a PPI and to help clinical decision-making regarding PPI use.

The strength of the present study was the long-term medication period of the participants, with 45.6% receiving a PPI for over a decade. Only a very few studies comprising patients who have received PPIs for >10 years have been conducted; therefore, our study will be helpful in understanding and assessing the safety and complications of long-term PPI treatment. However, this study also had some limitations. One limitation of this study is the small number of patients and the diversity of patient backgrounds, including *H. pylori* infection status and atrophic patterns. *H. pylori* infection induces hypergastrinemia by attacking somatostatin cells [37], and PPI-related endoscopic features differ depending on *H. pylori* infection status and eradication therapy [23,29,38,39,40]. The evaluation of *H. pylori* infection should be integrated into future studies. However, the diversity of this study did not affect our results because there were no significant changes in *H. pylori* infection status, atrophic patterns, or intestinal metaplasia among the study groups. The second limitation was the lack of histological evaluation for the diagnosis of atrophy and intestinal metaplasia as well as for FGPs and hyperplastic polyps. Finally, this was a cross-sectional study in which baseline gastrin levels and endoscopic findings before PPI use were unavailable.

Hypergastrinemia occurs in a variety of situations, including long-term administration of PPIs and AIG, so the gastric mucosa situation is different, and the details of the mechanisms by which a PPI affects gastrin mucosal changes are not clear, so further research is required. Furthermore, longitudinal studies involving larger patient cohorts are needed to assess the long-term safety of PPI use and its potential association with tumorigenic changes.

## 5. Conclusions

This study found no increase in the risk of gastric epithelial changes including tumor development, even after >10 years of PPI treatment. Gastrin levels in patients using a PPI for the long term positively correlated with CCLM development, regardless of the duration of PPI use; therefore, more detailed monitoring of gastrin levels and endoscopy, including gastric biopsy, is necessary in patients with CCLM.

## Figures and Tables

**Figure 1 diagnostics-14-02540-f001:**
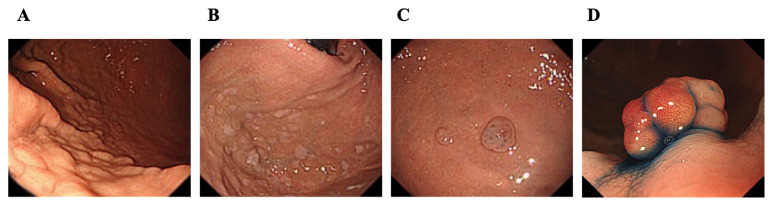
PPI-related endoscopic images: (**A**) cracked and cobblestone-like mucosa; (**B**) multiple white and flat elevated lesions; (**C**) fundic gland polyps and black spots; and (**D**) hyperplastic polyps.

**Figure 2 diagnostics-14-02540-f002:**
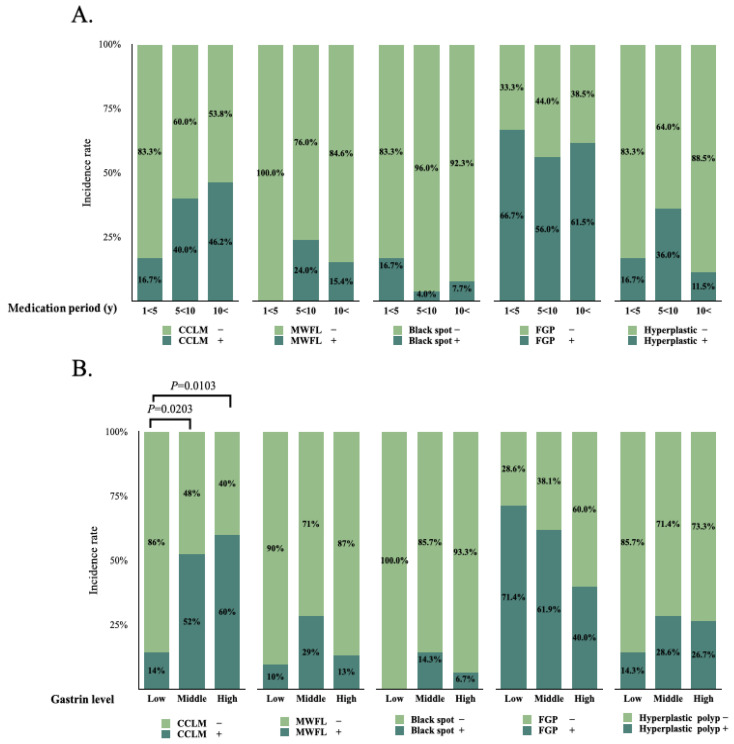
Incidence of PPI-related endoscopic gastric mucosal changes according to the medication period (years) (**A**) and gastrin levels (**B**). CCLM: cracked and cobblestone-like mucosa; MWFL: multiple white and flat elevated lesions; FGP: fundic gland polyp.

**Table 1 diagnostics-14-02540-t001:** Baseline characteristics and endoscopic findings.

Baseline Characteristics (*n* = 57)	No. (%)
Age, median (IQR)	73 (68.0–77.0)
Male sex, No. (%)	22 (38.6)
Smoking, No. (%)	6 (10.5)
Alcohol, No. (%)	18 (31.6)
*H. pylori* infection status, No. (%)	
No infection	29 (50.8)
Post eradication therapy	19 (33.3)
Current infection	6 (10.5)
Undefined	3 (5.3)
Atrophic pattern, No. (%)	
No atrophy	33 (57.9)
Closed-type	11 (19.3)
Open-type	13 (22.8)
Intestinal metaplasia, No. (%)	
0	47 (82.4)
1	4 (7.0)
2	6 (10.5)
Gastrin (pg/mL), median (IQR)	310 (190–510)
Duration of PPI use, No. (%)	
≥1, <5	6 (10.5)
≥5, <10	25 (43.9)
≥10	26 (45.6)
PPI-related endoscopic findings	
Cracked and cobblestone-like mucosa	23 (40.4)
Multiple white and flat elevated lesions	10 (17.5)
Black spots	4 (7.0)
Fundic gland polyps	34 (59.6)
Hyperplastic polyps	13 (17.5)

IQR: interquartile range.

**Table 2 diagnostics-14-02540-t002:** Patient characteristics classified according to PPI medication period.

Medication Period (Years)	≥1, <5 *n* = 6	≥5, <10 *n* = 25	≥10 *n* = 26	*p*-Value
Age, median (IQR)	71 (70–75)	74 (72–78)	72 (65.5–77)	0.452
Male sex, No. (%)	3 (50.0)	6 (24.0)	13 (50.0)	0.1449
Smoking, No. (%)	0 (8.3)	8 (32.0)	9 (34.6)	0.7062
Alcohol, No. (%)	2 (33.3)	8 (32.0)	9 (34.6)	1.0000
*H. pylori* infection status, No. (%)				
No infection	3 (50.0)	11 (44.0)	15 (57.7)	0.7414
Post eradication therapy	2 (33.3)	8 (32.0)	9 (34.6)	
Current infection	1 (16.7)	4 (16.0)	1 (3.8)	
Undefined	0 (0.0)	2 (8.0)	1 (3.8)	
Atrophic pattern, No. (%)				
No atrophy	3 (50)	13 (52.0)	17 (65.4)	0.7832
Closed-type	2 (33.3)	5 (20.0)	4 (15.4)	
Open-type	1 (16.7)	7 (28.0)	5 (19.2)	
Intestinal metaplasia, No. (%)				
0	5 (83.3)	20 (80.0)	22 (84.6)	0.749
1	1 (16.7)	2 (8.0)	1 (3.8)	
2	0 (0.0)	3 (12.0)	3 (11.5)	
Gastrin (pg/mL), median (IQR)	270 (140–333)	380 (220–610)	260 (180–365)	0.1205
ECM	0 (0)	2 (11.5)	3 (8.0)	1.0000
Parietal cell hyperplasia	1 (16.7)	15 (60)	12 (46.2)	0.1544

IQR: interquartile range.

**Table 3 diagnostics-14-02540-t003:** Relationship between the incidence of endoscopic findings and PPI medication period (years).

	OR	95% CI	*p*-Value
Cracked and cobblestone-like mucosa			
≥1, <5	1.00	-	-
≥5, <10	3.33	(0.33, 33.63)	0.307
≥10	4.29	(0.43, 42.81)	0.215
Multiple white and flat elevated lesions			
≥1, <5	NA	NA	NA
≥5, <10	NA	NA	NA
≥10	NA	NA	NA
Black spots			
≥1, <5	1.00	-	
≥5, <10	0.21	(0.01, 4.02)	0.299
≥10	0.42	(0.03, 5.66)	0.511
Fundic gland polyps			
≥1, <5	1.00	-	-
≥5, <10	0.64	(0.10, 4.21)	0.639
≥10	0.80	(0.12, 5.29)	0.817
Hyperplastic polyps			
≥1, <5	1.00	-	-
≥5, <10	2.81	(0.28, 28.55)	0.382
≥10	0.65	(0.05, 7.81)	0.736

OR: odds ratio; CI: confidence interval; NA: not applicable.

**Table 4 diagnostics-14-02540-t004:** Patient characteristics classified according to gastrin levels.

Gastrin Level (pg/mL)	Low <250 *n* = 21	Middle ≥250, <500 *n* = 21	High ≥500 *n* = 15	*p*-Value
Age, median (IQR)	72 (60–75)	72 (67–77)	75 (72.5–79)	0.051
Male sex, No. (%)	10 (47.6)	9 (42.9)	3 (20.0)	0.247
Smoking, No. (%)	3 (14.3)	2 (9.5)	1 (6.7)	0.872
Alcohol, No. (%)	4 (19.0)	12(57.1)	3(20.0)	0.021
Duration of anti-acid use, No. (%)				
≥1, <5	3 (14.3)	2 (9.5)	1 (6.7)	0.925
≥5, <10	8 (38.1)	9 (42.9)	8 (53.3)	
≥10	10 (47.6)	10 (47.6)	6 (40.0)	
*H. pylori* infection status, No. (%)				
No infection	13 (61.9)	9 (42.9)	7 (46.7)	0.292
Post eradication therapy	7 (33.3)	6 (28.6)	6 (40.0)	
Current infection	0 (0.0)	5 (23.8)	1 (6.7)	
Undefined	1 (4.8)	1 (4.8)	1 (6.7)	
Atrophic pattern, No. (%)				
No atrophy	14 (66.7)	12 (57.1)	7 (46.7)	0.337
Closed-type	5 (23.8)	4 (19.0)	2 (13.3)	
Open-type	2 (9.5)	5 (23.8)	6 (40.0)	
Intestinal metaplasia, No. (%)				
0	18 (85.7)	16 (76.2)	13 (86.7)	0.724
1	2 (9.5)	2 (9.5)	0 (0.0)	
2	1 (4.8)	3 (14.3)	2 (13.3)	
ECM	2 (9.5)	1 (4.8)	2 (13.3)	0.842
Parietal cell hyperplasia	5 (23.8)	12 (57.1)	11 (73.3)	0.009

**Table 5 diagnostics-14-02540-t005:** Relationship between endoscopic findings and gastrin levels.

	OR	95% CI	*p*-Value	P for Trend
Cracked and cobblestone-like mucosa				
Low	1.00	-	-	0.0171
Middle	6.60	(1.46, 29.75)	0.014	
High	9.00	(1.79, 45.23)	0.008	
Multiple white and flat elevated lesions				
Low	1.00	-	-	0.2668
Middle	3.80	(0.66, 21.93)	0.136	
High	1.46	(0.17, 11.95)	0.723	
Black spots				
Low	NA	NA	NA	NA
Middle	NA	NA	NA	NA
High	NA	NA	NA	NA
Fundic gland polyps				
Low	1.00	-	-	0.1784
Middle	0.65	(0.18, 2.40)	0.517	
High	0.27	(0.06, 1.10)	0.067	
Hyperplastic polyps				
Low	1.00	-	-	0.5169
Middle	2.40	(0.50, 11.42)	0.271	
High	1.88	(0.40, 11.82)	0.365	

OR: odds ratio; CI: confidence interval; NA: not applicable.

**Table 6 diagnostics-14-02540-t006:** Relationship between the incidence of cracked and cobblestone-like mucosa and gastrin levels after adjusting for potential confounders.

	OR	95% CI	*p*-Value	P for Trend
Age-adjusted model				
Low	1.00	-	-	
Middle	8.27	(1.58, 43.27)	0.012	0.0158
High	13.20	(2.12, 82.13)	0.006	
Sex-adjusted model				
Low	1.00	-	-	
Middle	8.32	(1.39, 49.75)	0.020	0.0269
High	15.69	(2.04, 120.97)	0.008	
Period-adjusted model				
Low	1.00	-	-	
Middle	6.84	(1.47, 31.86)	0.014	0.0130
High	9.72	(1.98, 47.59)	0.005	
*H. pylori*-adjusted model				
Low	1.00	-	-	
Middle	7.63	(1.51, 38.67)	0.014	0.0118
High	10.69	(2.07, 55.19)	0.005	
Atrophy-adjusted model				
Low	1.00	-	-	
Middle	7.42	(1.64, 33.48)	0.009	0.0058
High	11.36	(2.39, 54.03)	0.002	
IM-adjusted model				
Low	1.00	-	-	
Middle	7.29	(1.59, 33.34)	0.010	0.0136
High	9.34	(1.85, 47.01)	0.007	

OR: odds ratio; CI: confidence interval; IM: intestinal metaplasia.

## Data Availability

Raw data were generated at the Hattori Clinic. Derived data supporting the findings of this study are available from the corresponding author upon request.

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
