# Peer review of "Effects of Medication Period and Gastrin Levels on Endoscopic Gastric Mucosal Changes in Long-Term Proton Pump Inhibitor Users"

_diagnostics, 2024, doi:10.3390/diagnostics14222540_

Round 1
Reviewer 1 Report
Comments and Suggestions for Authors
I have carefully and devotedly read the manuscript entitled: "Effects of medication period and gastrin levels on endoscopic gastric mucosal changes in long-term proton pump inhibitor users", which showed that long-term exposure to PPi use is not associated with an increased incidence of gastric lesions. I believe that the manuscript is worthy of publication prior to some major revisions:
- The main limitation of the manuscript, as it is metologically described to date, is the variability in the timing of EGDs versus the duration of PPi therapy: the duration of PPi therapy at the time of gastroscopy is not known, so data on the influence of PPi on retrospectively reviewed endoscopic findings cannot be accurately inferred. If it is not possible to change the study design, as I imagine, at least it is mandatory to make explicit the time of duration of therapy not at the time of enrollment but at the time of index gastroscopy, in order to have at least indicative data of possible correlation between time of exposure to therapy and endoscopic findings.
- As briefly mentioned within the study's limitations, the variables influencing hypergastrinemia in the population under examination are multiple: exposure to PPI therapy, the possibility of including patients with autoimmune atrophic chronic gastritis (as inferred from data related to atrophy class "O"), and cases with current or past H. pylori infection. Therefore, the population under examination is considered "heterogeneous," and it is not possible to strictly deduce data regarding the sole causality of PPIs concerning endoscopic findings: please consider revising data including only patients with no other causes of hypergastrinemia than PPi therapy ongoing.
- Hypergastrinemia represents a driver of gastric mucosal proliferation and also of aberrant proliferation of ECL cells, leading to an increased risk of gastric neuroendocrine neoplasms. This pathophysiological mechanism is not clearly outlined in the introduction of the manuscript, and neither the methods nor the results provide data regarding the possible presence of gastric neuroendocrine neoplasms (gNENs) nor the description of the types and prevalence of ECL hyperplasia observed (linear, micronodular, macronodular). Although not numerically significant in the population under study, I request that these details be thoroughly included in the manuscript.
- How did the authors explain themselves the results showing an increased prevalence of CCLM in patients with higher gastrin levels? A deeper insight into the possible pathophysiological rationale is requested.
- I believe it is overly provocative to state that, based on circulating gastrin levels at the time of the index gastroscopy in patients indicated for proton pump inhibitor therapy, a targeted therapeutic choice can be made. This is especially because the collected data do not support this hypothesis (circulating gastrin levels are measured after the EGDs in the enrolled patients) and because evidence regarding fluctuations in circulating gastrin levels in patients with a long history of pCABs exposure has not yet been generated. Please reformulate.
- Some relevant studies on hypergastrinemia have not been included, and I suggest adding them to the bibliography: DOI 10.1016/j.dld.2023.02.008, DOI 10.1016/j.dld.2024.07.024.
Reviewer 2 Report
Comments and Suggestions for Authors
Manuscript entitled "Effects of medication period and gastrin levels on endoscopic gastric mucosal changes in long-term proton pump inhibitor users" by Hiroko Suda, et al.
This manuscript provides an insightful analysis of the impact of long-term proton pump inhibitor (PPI) use on gastric mucosal changes and the association with gastrin levels. The study addresses an important topic as the global usage of PPIs has increased, with significant clinical implications. The authors have conducted a solid investigation, which adds valuable data to the existing literature.
Comments:
1. The Introduction section offers a good overview of the background of PPI usage. However, it would be beneficial to further elaborate on the pathophysiology underlying the gastric mucosal changes seen with PPI use, particularly focusing on the mechanisms of hypergastrinemia and its long-term effects on gastric tissue. Additionally, a clearer articulation of the study's novelty compared to prior research could strengthen the rationale.
2. The Inclusion and Exclusion Criteria are clearly presented; however, more explanation regarding the exclusion of patients with poor compliance (<90%) and those with certain comorbidities would help clarify how these exclusions impacted the study population. Additionally, it would be beneficial to specify whether any other medications that could influence gastric physiology were controlled for during the selection of participants.
3. In terms of baseline characteristics, the manuscript reports that there were no statistically significant differences across the groups. However, a more detailed discussion on the specific variables assessed (such as age, gender, smoking habits, alcohol consumption, etc.) and how these were accounted for in the statistical analysis would enhance the readers' understanding of the homogeneity of the study population. Including p-values in Table 1 to explicitly show the lack of significant differences between groups would also be valuable.
4. The Methods are described in a clear and structured manner. However, some methodological details require further clarification. For instance, the process for stratifying patients based on gastrin levels should be better explained, particularly how the threshold values for low, middle, and high gastrin groups were determined. This will help readers understand the clinical relevance of these cut-offs.
5. The presentation of the results is well-organized. However, the Table 1 listing baseline characteristics could benefit from clearer headings and a more intuitive layout, especially concerning the categorization of PPI usage and its impact on endoscopic findings. Also, the authors should consider including a graphical representation of the relationship between gastrin levels and the risk of mucosal changes, as this would visually enhance the interpretation of the data.
6. The statistical analysis section is robust; however, the limitations due to the small sample size should be addressed more explicitly in the Discussion. It is important to acknowledge how this might impact the generalizability of the findings. Additionally, a more thorough comparison with existing studies on PPI-related mucosal changes would enrich the discussion and highlight the contributions of this research more effectively.
7. One notable omission is the discussion of knowledge gaps and future research directions. While the study offers valuable findings, it would benefit from a more explicit description of existing knowledge gaps in the field, such as the unclear mechanisms of PPI-induced mucosal changes over time and how hypergastrinemia might differentially affect various patient populations. Furthermore, the authors should suggest future research avenues, such as longitudinal studies involving larger patient cohorts to assess the long-term safety of PPI use and its potential links to neoplastic changes.
8. Similarly, the clinical implications and recommendations are somewhat underdeveloped. Although the results highlight a correlation between high gastrin levels and the development of certain gastric mucosal changes, the authors should offer more detailed recommendations for clinical practice. For instance, guidelines for monitoring gastrin levels in long-term PPI users, or criteria for considering de-prescription of PPIs in patients with elevated gastrin, would provide useful guidance for practitioners. Additionally, discussing potential risk stratification strategies based on gastrin levels could offer practical value to the study's findings.
9. Lastly, the conclusion is sound but could benefit from a more forward-looking perspective. For instance, discussing potential clinical recommendations for monitoring patients on long-term PPIs based on their gastrin levels would provide practical relevance to the study's findings.
Round 2
Reviewer 1 Report
Comments and Suggestions for Authors
I thank the Authors for providing un updated and more precise version of the manuscirpt, which I believe is now acceptable for publication.
Reviewer 2 Report
Comments and Suggestions for Authors
The authors have addressed all of my concerns satisfactorily, and the manuscript can now be accepted for publication.